# Renal Endocannabinoid Dysregulation in Obesity-Induced Chronic Kidney Disease in Humans

**DOI:** 10.3390/ijms241713636

**Published:** 2023-09-04

**Authors:** Anna Permyakova, Ariel Rothner, Sarah Knapp, Alina Nemirovski, Danny Ben-Zvi, Joseph Tam

**Affiliations:** 1Obesity and Metabolism Laboratory, The Institute for Drug Research, School of Pharmacy, Faculty of Medicine, The Hebrew University of Jerusalem, Jerusalem 9112001, Israel; anna.permyakova@mail.huji.ac.il (A.P.); ariel.rothner@mail.huji.ac.il (A.R.); alina.nemirovskai@mail.huji.ac.il (A.N.); 2Department of Developmental Biology and Cancer Research, Institute for Medical Research Israel Canada, Hadassah Medical School–The Hebrew University of Jerusalem, Jerusalem 9112001, Israel; sarah.knapp@mail.huji.ac.il (S.K.); danny.ben-zvi@mail.huji.ac.il (D.B.-Z.)

**Keywords:** obesity, endocannabinoid system, chronic kidney disease

## Abstract

The endocannabinoid system (ECS) regulates various physiological processes, including energy homeostasis and kidney function. ECS upregulation in obese animals and humans suggests a potential link to obesity-induced chronic kidney disease (CKD). However, obesity-induced ECS changes in the kidney are mainly studied in rodents, leaving the impact on obese humans unknown. In this study, a total of 21 lean and obese males (38–71 years) underwent a kidney biopsy. Biochemical analysis, histology, and endocannabinoid (eCB) assessment were performed on kidney tissue and blood samples. Correlations between different parameters were evaluated using a comprehensive matrix. The obese group exhibited kidney damage, reflected in morphological changes, and elevated kidney injury and fibrotic markers. While serum eCB levels were similar between the lean and obese groups, kidney eCB analysis revealed higher anandamide in obese patients. Obese individuals also exhibited reduced expression of cannabinoid-1 receptor (CB1R) in the kidney, along with increased activity of eCB synthesizing and degrading enzymes. Correlation analysis highlighted connections between renal eCBs, kidney injury markers, obesity, and related pathologies. In summary, this study investigates obesity’s impact on renal eCB “tone” in humans, providing insights into the ECS’s role in obesity-induced CKD. Our findings enhance the understanding of the intricate interplay among obesity, the ECS, and kidney function.

## 1. Introduction

Obesity has emerged as a global public health issue, characterized by a rising prevalence of overweight and obese individuals. The World Health Organization reports that over 1 billion people worldwide are presently classified as obese, comprising 650 million adults, 340 million adolescents, and 39 million children. Notably, the prevalence of obesity has nearly tripled since 1975. This surge in body weight is intricately linked to the development of various chronic health conditions, including chronic kidney disease (CKD [1,2,3]).

CKD is a gradually deteriorating condition characterized by a decline in renal function, and it is estimated that approximately 10% of the global population is affected by CKD [4,5]. The progression of CKD can eventually lead to end-stage kidney disease (ESKD), necessitating renal replacement therapies such as dialysis or kidney transplantation. ESKD significantly impairs quality of life and is associated with a high mortality rate [6,7,8]. The association between obesity and CKD is intricate and multifaceted, involving a variety of mechanisms. These mechanisms include increased glomerular pressure, inflammation, oxidative stress, metabolic abnormalities, adipokines, and the involvement of the endocannabinoid system (ECS) [9,10,11]. Understanding the interplay between these factors is crucial for comprehending the pathophysiological pathways connecting obesity and CKD.

The ECS is a highly complex signaling system comprising endocannabinoids (eCBs), cannabinoid-1 and -2 receptors (CB1R and CB2R, respectively), and enzymes responsible for eCB synthesis and degradation. All these elements play vital roles in various physiological and pathological processes, including appetite regulation, energy metabolism, and inflammation. The two primary eCBs, anandamide (AEA or *N*-arachidonoylethanolamine) and 2-arachidonoylglycerol (2-AG), interact with CB1R and CB2R, which are expressed in diverse tissues, including the kidney [12,13]. Additionally, this network encompasses eCB-like compounds, *N*-oleoylethanolamine (OEA) and *N*-palmitoylethanolamine (PEA), which share their catabolic pathway with AEA [14,15,16,17].

Studies have consistently demonstrated ECS alterations within the kidney of both obese individuals and animal models of obesity. Specifically, obese animals have exhibited increased levels of AEA and 2-AG within their kidneys [18,19,20], which have been associated with renal dysfunction. Furthermore, research conducted by our group and others has revealed significant changes in the expression of CB1R in whole kidney tissue [20,21,22], glomeruli [23,24,25], and proximal tubular cells [20,26] in the context of obesity. These alterations have been linked to intracellular lipid accumulation, inflammation, fibrosis, and kidney injury [20,27]. Notably, studies utilizing deletion or inhibition of CB1R in obese animals have demonstrated improvements in kidney function [20,21,24,25,28,29,30,31,32]. Despite these findings, there remains a dearth of information regarding the involvement of ECS in obesity-induced CKD, particularly in humans. Thus, the objective of this study is to contribute to the existing knowledge on this subject matter by investigating the impact of obesity on the ECS in the kidneys of humans and gain insights into the role of the ECS in obesity-induced CKD.

## 2. Results

### 2.1. Patient Information

Patient demographics and key parameters are specified in Table 1. The study comprised 21 male patients who underwent surgery for the removal of a kidney lesion, with 10 individuals classified as having a low body mass index (BMI; <26) and 11 individuals classified as having a high BMI (>30). Lesion characterization revealed that 43% of participants had clear cell renal cell carcinoma, 24% had papillary renal cell carcinoma, and the remaining patients did not have a malignant tumor. No significant association was observed between BMI and lesion characterization or malignancy.

For further investigation, serum samples and healthy tissue sections from the lesion were utilized. Fasting serum glucose levels were available for most patients from their medical records. Biochemical measurements, including creatinine, blood urea nitrogen (BUN), glucose, lactate, alanine transaminase (ALT), aspartate aminotransferase (AST), alkaline phosphatase (ALP), triglycerides (TG), cholesterol, high-density lipoprotein (HDL), and low-density lipoprotein (LDL), were conducted using the serum samples. Interestingly, the low- and high-BMI groups exhibited similar biochemical profiles, with no significant differences observed.

### 2.2. Patients with High BMI Exhibit Abnormal Kidney Morphology and Elevated Kidney Injury Markers

Obesity is known to contribute to CKD and is associated with various structural and functional abnormalities in the kidney. To assess the morphological changes in the kidneys of obese patients, kidney tissues were subjected to H&E staining. Figure 1 presents representative microscope images at 20× magnification (Figure 1A) and the quantification of the glomerular area (Figure 1B) and Bowman’s space area (Figure 1C). While no significant differences were observed in glomerular size between the lean and the obese groups, obese patients exhibited a significant enlargement of Bowman’s space, as indicated by the quantification analysis (*p* = 0.0317). Furthermore, a significant increase in kidney injury molecule-1 (KIM-1) protein expression in the kidneys of obese patients compared to the lean group (*p* = 0.0357) was found (Figure 1D,E). Additionally, a significant upregulation in the gene expression levels of transforming growth factor beta (*TGFΒ*) and interleukin 18 (*IL-18*) (Figure 1F), which are associated with inflammation and fibrosis, were found in high-BMI individuals compared to the lean group (*p* = 0.0006 and *p* = 0.008, respectively). Additionally, we attempted to measure other genes associated with inflammation and fibrosis, such as tumor necrosis factor alpha (*TNFα*), interleukin 6 (*IL-6*), interferon gamma-induced protein 10 (*IP-10*), TIMP metallopeptidase inhibitor 1 (*TIMP1*), fibronectin (*FN*), and collagen I (*COL1*); however, none of them changed in the obese patients compared to the lean controls (Figure A1). Nevertheless, trichrome staining of kidney sections revealed elevated levels of collagen in the obese group, further suggesting increased fibrosis (Figure 1G,H). Taken together, these results suggest a higher propensity for CKD progression in the obese population compared to their lean counterparts.

### 2.3. AEA Levels Are Increased in the Kidneys of Obese Patients

Obesity has been linked to an increase in renal eCB “tone” in animal studies [20]. However, this phenomenon has not been thoroughly examined in humans. Using LC-MS/MS, we measured the concentrations of 2-AG, AEA, and their related ligands, OEA and PEA, as well as the eCB degrading product arachidonic acid (AA), in the serum and kidney of obese and lean patients. Interestingly, none of the measurements showed any significant change in the serum of obese individuals (Figure 2A–E). In renal eCB “tone”, we found a significant increase in the AEA levels of obese patients (*p* = 0.0401), together with a noticeable but non-significant trend toward elevated levels of AA, the degrading product of AEA (*p* = 0.1206) (Figure 2F–J). In addition, a slight non-significant reduction in renal 2-AG was found in the obese patients. These findings provide initial evidence suggesting that obesity may contribute to alterations in the renal ECS in humans and highlight its potential in the pathophysiology of obesity-related renal dysfunction.

### 2.4. Abnormal Expression of CB1R and ECS-Related Enzymes in Obese Patients

Previous research has indicated alterations in the renal expression of CB1R and CB2R in both humans and animals under lipotoxic conditions [32,33,34,35]. Building upon the findings presented in Figure 2, we sought to further investigate the expression levels of CB1R. Figure 3 illustrates the downregulation of CB1R expression in the obese group (Figure 3A), as depicted by the representative blot and quantification (*p* = 0.0021). As we found changes in renal eCB abundance amongst the groups, we further analyzed the expression levels of their metabolic machinery. The expression levels of fatty acid amide hydrolase (FAAH), the catabolic enzyme of *N*-acylethanolamines (AEA, OEA, and PEA), were higher in the obese patients, in both the protein (*p* = 0.119; Figure 3B) and the mRNA (*p* = 0.0205; Figure 3C) expression levels. Moreover, the obese patients also displayed increased gene expression levels of *N*-acylphosphatidylethanolamine phospholipase D (*NAPEPLD*; *p* = 0.0093), the *N*-acylethanolamine anabolic enzyme. Finally, the gene expression of the synthesizing and degrading enzymes of 2-AG, diacyl-glycerol lipases α and β (*DAGLA* and *DAGLB*; *p* = 0.0401 and 0.0076, respectively) and monoacylglycerol lipase (*MGLL*; 0.0111), respectively, was also notably higher in the obese individuals (Figure 3C). These findings provide valuable insights into the dysregulation of the ECS in the context of obesity-related renal pathophysiology.

### 2.5. Multiple-Parameter Correlations

In order to gain deeper insights into the relationship between obesity-induced CKD and changes in ECS “tone”, we conducted a comprehensive analysis using a multiple-parameter correlation matrix, as presented in Figure 4 (and explained in detail in Section 4). In addition, to compare the kidney injury markers with ECS parameters, we chose to include other variables known to be associated with CKD in a more indirect manner, such as liver function parameters and glucose and lipid profiles, which have been shown repeatedly to affect kidney function [36,37,38,39,40,41,42,43]. Initially, when considering both lean and obese groups together, we observed minimal correlations between renal and circulating eCBs, with few noticeable associations (Figure 4A). However, upon stratifying the data into lean (Figure 4B) and obese (Figure 4C) groups, a multitude of previously unrecognized connections emerged, even though not all of them were statistically significant.

In obese patients, we observed a positive correlation between KIM-1 and BMI, while their correlation with renal and circulating eCBs predominantly showed negative associations. Furthermore, histopathological parameters of kidney function displayed predominantly negative correlations with both renal and circulating eCBs in obese patients, with a notable exception of a strong positive correlation with renal AEA. Conversely, in lean patients, the correlations with renal eCBs were consistently negative, except for positive associations with circulating OEA and AEA. Additionally, serum levels of BUN and creatinine exhibited negative correlations with kidney AEA and positive correlations with kidney PEA and OEA in the high-BMI group. In the low-BMI group, the only notable correlation was a positive association with renal 2-AG. Furthermore, we observed a clear and robust association between renal eCBs, circulating 2-AG, and serum glucose levels, as well as liver and lipid profiles, in both lean and obese patients. These connections underscore the interplay between renal eCBs and metabolic parameters in both groups (Table A1).

Overall, our findings suggest distinct patterns of correlations between renal and circulating eCBs, as well as their associations with clinical parameters, depending on the BMI status of the patients. This comprehensive analysis provides novel insights into the complex relationship between obesity-induced CKD and the ECS, shedding light on potential mechanisms underlying the disease progression. Further investigations are warranted to unravel the functional implications of these correlations and their relevance for targeted therapeutic interventions.

## 3. Discussion

The ECS has been implicated in the pathogenesis of obesity-induced CKD due to its influence on metabolism and renal function [9,10,11]. However, the precise mechanisms underlying the ECS’s involvement in CKD, particularly in humans, remain poorly understood. This study aimed to enhance our understanding by analyzing a small cohort of human participants.

Our findings demonstrated evidence of kidney injury in obese patients, as indicated by histological staining and the presence of kidney injury markers. Furthermore, we investigated the levels of eCBs extracted from the kidneys of lean and obese patients, providing novel insights. Specifically, we observed elevated levels of AEA and a slight non-significant reduction in 2-AG in the kidneys of obese individuals. Additionally, we observed a significant decrease in kidney CB1R expression levels and an upregulation of eCB biosynthesis and degradation enzymes in the obese group. Notably, there were no significant alterations in circulating eCBs between the lean and obese groups. Moreover, we employed a comprehensive correlation matrix analysis to elucidate the relationship between kidney eCB “tone” and the overall metabolic state. This large-scale analysis highlighted the intricate interplay between eCBs and systemic metabolic factors. While our study suggests a role of the ECS in the pathogenesis of CKD, further investigations are required to fully elucidate the underlying mechanisms.

The serum biochemistry of obese individuals in our study did not exhibit significant differences compared to their lean counterparts. However, assessment of renal histopathology and injury markers revealed evidence of more severe renal pathology in the obese group. Enlarged glomerular area and Bowman’s space are well-established indicators of kidney injury in obesity-induced CKD [44,45]. Previous studies in obese human patients have consistently reported these findings [46,47,48]. Consistent with the existing literature, our histopathological analysis also demonstrated a noteworthy expansion of Bowman’s space in the kidney samples of the obese participants. Although our cohort was limited in size and did not demonstrate a statistically significant glomerular enlargement, our results are aligned with the literature on kidney injury in obesity-induced CKD. Furthermore, we observed increased levels of KIM-1 protein and *TGFB* and *IL-18* mRNA expression, which are additional markers of kidney injury [49,50,51,52,53,54,55,56]. Trichrome staining for collagen has also indicated increased fibrosis in the high-BMI group, further verifying the development of CKD [57]. Our multi-parameter correlation analysis revealed a strong positive correlation between KIM-1 and BMI in obese patients, thus linking kidney injury to the increased body weight of the subjects.

This study represents the first investigation of eCBs within the renal context of lean and obese human patients. Notably, obese individuals display significantly elevated AEA levels within their kidneys. These findings are in line with previous studies involving both in-vivo and in-vitro models of obesity, which have highlighted an elevated kidney eCB “tone” [18,19,20], as evidenced by increased levels of AEA and/or 2-AG in response to high-fat diet and lipotoxic conditions [20,26]. Concomitant with these eCB changes, a substantial overexpression of renal CB1R has been reported in obese mice [20,28]. This elevation in ECS “tone” and the resultant CB1R overactivation play contributory roles in the onset of kidney damage, inflammatory responses, and lipid accumulation [20], with pharmacological blockade and genetic deletion of renal CB1R shown to mitigate the damage [20,22,28]. Furthermore, CB1R overactivity in lipotoxic conditions also contributes to mitochondrial dysfunction [58], endoplasmic reticulum stress, and apoptosis in renal proximal tubule cells [28]. Surprisingly, we observed a significant downregulation of renal CB1R protein expression in the obese subjects, contrary to previous in-vivo and human studies, including some conducted by our group [20,22,28,59]. A possible explanation for these discrepancies is the inability to control the sample location within the kidney (such as cortex or medulla) due to the dependency on the location of the tumor site, and the lack of provided data regarding this aspect. Additionally, a limitation of the current study is the inability to capture the dynamic changes in ECS activity at different time points. Nevertheless, we propose that the observed downregulation of CB1R may be a compensatory mechanism employed by the organism in response to mitigate the detrimental effects of obesity on the kidney. This downregulation of CB1R assumes significance, particularly in the context of emerging therapies targeting renal CB1R as a potential intervention for CKD with peripherally restricted CB1R blockers [21,22,26,29,33]. Furthermore, our findings indicate that all the enzymes involved in eCB biosynthesis and degradation exhibit elevated mRNA levels in our cohort of high-BMI patients, introducing complexity into the interpretation of these results and requiring further investigation. Altogether, the activity of the ECS undergoes alterations under obese conditions; however, the direction of these changes remains unclear, at least in humans.

In the context of this study, it was observed that among the obese patients examined here, there were elevated renal AEA levels. Conversely, the levels of 2-AG did not exhibit significant differences and displayed a marginal decreasing trend. It is worth mentioning that although the renal levels of 2-AG exceeded those of AEA, the question of which eCB is more abundantly and biologically effective within the kidney remains a topic of debate. This discrepancy in abundance is known to vary across distinct compartments of the kidney [33]. Furthermore, it is important to consider that these eCBs exhibit different affinities and functional activities. AEA functions as a high-affinity, CB1R-selective partial agonist, while 2-AG acts as a moderate-affinity agonist for both CB1R and CB2R [60]. This distinct pharmacological profile contributes to the intricate interplay between these endogenous compounds. Given these pharmacological distinctions and the distinct renal patterns observed between 2-AG and AEA, the implications for CB1R activation in the context of obesity remain to be further investigated. Moreover, while we found changes in renal eCB levels, we did not observe any differences in the circulating eCB levels between the two groups, which contrasts with previous reports of elevated serum AEA, OEA, and PEA in obese individuals [61,62,63,64,65]. This discrepancy may be attributed to the limited size of our cohort or the lack of control over sample timing, as various physiological factors are known to influence circulating eCB levels [66,67,68,69]. Notably, our study found no correlations between the renal and circulating eCB levels, suggesting their distinct roles as local regulators within the kidney.

Our study employed a comprehensive correlation matrix analysis to explore the relationship between renal eCB levels, kidney injury parameters, and kidney and systemic health markers. Initially, weak negative correlations were observed between histological kidney injury parameters and renal eCB levels when considering all patients collectively. However, when stratifying the data based on obesity status, interesting patterns emerged. In lean individuals, serum kidney function markers (creatinine and BUN) exhibited a positive correlation solely with renal 2-AG. Conversely, in obese individuals, these markers displayed stronger correlations with AEA, OEA, and PEA. Notably, this connection between kidney health markers and renal *N*-acylethanolamines was reinforced by their robust correlations with renal histological damage markers in the obese group. In contrast, in the lean group, renal AEA, OEA, and PEA were negatively correlated with histopathological markers, indicating a divergent relationship compared to the obese group. Furthermore, we observed a prominent negative correlation between histological markers and renal 2-AG and AA in the obese group, inconsistent with previous findings in the kidneys of obese animals [18,19,20]. Regarding AEA, there are conflicting reports in the literature regarding its role in the kidney [11,33,70], although numerous studies on chronic renal conditions describe elevated renal AEA levels [18,20,71,72]. Overall, our findings highlight a distinct and contrasting profile in renal eCBs and their correlation with kidney health markers between lean and obese patients. This underscores the potential influence of obesity on the renal ECS and its intricate interplay with kidney function and pathology.

In addition to the established association between the ECS and kidney health, our large-scale correlation analysis revealed significant connections between renal ECS and systemic metabolic parameters. Notably, renal eCB levels showed a close relationship with blood glucose levels. In lean patients, AEA, OEA, PEA, and AA exhibited positive correlations with blood glucose levels. However, in the obese group, while still positively correlated with AEA, there were negative correlations observed with 2-AG, AA, and OEA. Existing in-vivo studies conducted by other research groups have reported upregulation of renal AEA in high-glucose environments, although evidence regarding 2-AG remains conflicting [18,34,73].

An intriguing and unexpected finding of our study is the strong dependence of renal “endocannabidiome” on lipid and liver profiles. In lean patients, ALT, AST, serum TG levels, and total cholesterol exhibited positive correlation with all eCBs, except for 2-AG, which displayed positive correlations with HDL and LDL. Conversely, in the obese group, all eCBs, except AEA, demonstrated positive correlations with ALT, AST, and cholesterol, while displaying negative correlations with ALP and TG. Notably, AEA exhibited an opposite pattern compared to the other eCBs across these parameters. Liver problems and dyslipidemia are common manifestations of the metabolic syndrome in obese patients [35,74,75,76], and previous animal studies, as mentioned earlier, have reported changes in renal eCBs in the context of obesity. However, our study is the first to establish correlations between lipid and liver profiles and the renal “endocannabidiome”.

Several notable limitations are inherent in the framework of this study. These encompass a relatively small sample size and the exclusion of female subjects, for example. In addition, even though we do possess some background knowledge about the comorbidities, as well as treatment regimens, of these patients, which potentially could affect the conclusions drawn from our results, we could not at this time investigate any such connections. Unfortunately, the comorbidities and treatment parameters are too varied, which makes it challenging to successfully correlate them with such a limited cohort of patients. Additionally, the utilization of biobank-derived specimens in this study as the basis for analysis impeded the meticulous control of pivotal factors such as temporal alignment of samples, spatial localization within the kidney, and controlled experimental conditions, which altogether are especially important for evaluating systemic eCBs profiling. Further prospective research with a larger sample size and more diverse patient population is warranted to validate our findings. Importantly, while this study aimed to evaluate the renal ECS changes in the obese population, the groups were determined by BMI, which is known to be a limited parameter in assessing metabolic health [77]. Further, while we did observe worsened kidney architecture and elevated injury markers in the obese group, it did not translate to robust functional damage. How ECS dysregulation in these obese patients affects their susceptibility to developing CKD and whether their ECS profile changes as renal function declines requires further research. Altogether, these findings provide clinical evidence of ECS dysregulation in the kidneys of obese patients, with implications for therapeutic targeting for obesity-induced CKD.

## 4. Materials and Methods

### 4.1. Study Population

The study comprised 21 male patients, with 10 classified as lean and 11 as obese, ranging in age from 38 to 71 years. These patients underwent tissue biopsies specifically for localized renal mass, with only the healthy tissue section utilized for subsequent analysis. Demographic information and comorbidities were obtained from the patients’ medical records. BMI, calculated as weight divided by height squared (kg/m^2^), was determined, and individuals with a BMI exceeding 30 were considered obese.

### 4.2. Study Protocol

The biological samples in this study, including frozen kidney tissue, frozen serum, and FFPE sections, were obtained from “MIDGAM”—Israel National Biobank for Research. This nonprofit organization, operating under the supervision of the Ministry of Health, serves as a facilitator for biomedical research and industry in Israel. The MIDGAM biobank collects samples from various donors, including patients with malignant and non-malignant diseases, as well as healthy volunteers. Renal tissues that were surgically removed or biopsied, as well as samples from blood, were collected, along with relevant demographic and clinical data, all in accordance with a legally approved protocol. All clinical data utilized in this study were obtained with approval from the Hadassah Medical School Institutional Review Board (IRB), with approval number HMO-0611-17, or Ministry of Health (MOH) IRB, with the assigned approval number 20185829. Informed consent was obtained from all patients involved, either through IRB approval at Hadassah Medical Center or by a waiver of consent for samples obtained from MIDGAM.

### 4.3. Biochemistry Measurements

The levels of various biochemical markers were measured to assess relevant parameters. Serum samples were used to determine the concentrations of creatinine, urea, glucose, lactate, ALT, AST, ALP, TG, total cholesterol, LDL, and HDL. These measurements were performed using a Cobas C-111 chemistry analyzer (Roche, Basel, Switzerland). BUN levels were calculated based on the serum urea concentrations (BUN mg/dL  =  Urea mM × 2.801). Fasting serum glucose levels were obtained from the patients’ medical records.

### 4.4. Endocannabinoid Extraction and Measurement by LC-MS/MS

The extraction, purification, and quantification of serum and kidney eCBs were performed using stable isotope dilution liquid chromatography/tandem mass spectrometry (LC-MS/MS) as previously described [73]. In brief, serum and kidney proteins were precipitated using ice-cold acetone and Tris buffer (50 mM, pH 8.0). Subsequently, an ice-cold extraction buffer (1:1 MeOH/Tris Buffer + an internal standard [d_4_-AEA]) was added to the samples. The homogenates were then extracted using a mixture of ice-col CHCl_3_:MeOH (2:1, vol/vol), followed by three washes with ice-cold chloroform. The samples were then dried under a nitrogen stream and reconstituted in MeOH.

The analysis by LC-MS/MS was performed using an AB Sciex (Framingham, MA, USA) QTRAP^®^ 6500+ mass spectrometer coupled with a Shimadzu (Kyoto, Japan) UHPLC System. A Kinetex 2.6 µm C18 (100 × 2.1 mm) column from Phenomenex (Torrance, CA, USA) was used for liquid chromatographic separation. The autosampler temperature was set at 4 °C, and the column was maintained at 40 °C throughout the analysis. Gradient elution mobile phases consisted of 0.1% formic acid in water (phase A) and 0.1% formic acid in acetonitrile (phase B).

eCBs were detected in a positive ion mode using electron spray ionization (ESI) and the multiple reaction monitoring (MRM) mode of acquisition. The collision energy (CE), declustering potential (DP), and collision cell exit potential (CXP) for the monitored transitions are given in Table 2. The levels of AEA, 2-AG, OEA, PEA, and AA in samples were measured against standard curves, which were then calculated in pmol/mL serum or pmol/mg kidney weight.

### 4.5. Real-Time PCR

Kidney mRNA was extracted using a Bio-Tri RNA lysis buffer (Bio-Lab, Jerusalem, Israel). Subsequently, DNase I treatment (Thermo Scientific, Hanover Park, IL, USA) was performed to remove any residual genomic DNA. The RNA samples were then reverse-transcribed using the qScript cDNA Synthesis kit (Quantabio, Beverly, MA, USA) to generate cDNA. Real-time PCR analysis was conducted using iTaq Universal SYBR Green Supermix (Bio-Rad, Hercules, CA, USA) and the CFX connect ST system (Bio-Rad, Hercules, CA, USA). The following primer pairs were utilized for amplification: *TGFB* (5′-CCCAGCATCTGCAAAGCTC-3′, 5′-GTCAATGTACAGCTGCCGCA-3′), *IL-18* (5′-GCCTAGAGGTATGGCTGTAA-3′, 5′-GCGTCACTACACTCAGCTAA-3′), *TNFα* (5′-GGTGCTTGTTCCTCAGCCTC-3′, 5′-CAGGCAGAAGAGCGTGGTG-3′), *IL-6* (5′-TAGCCGCCCCACAGACAG-3′, 5′-GGCTGGCATTTGTGGTTGGG-3′), *IP-10* (5′-GCCTAGAGGTATGGCTGTAA-3′, 5′-GCGTCACTACACTCAGCTAA-3′), *TIMP1* (5′-CTTCTGCAATTCCGACCTCGT-3′, 5′-ACGCTGGTATAAGGTGGTCTG-3′), *FN* (5′-CCACCCCCATAAGGCATAGG-3′, 5′-GTAGGGGTCAAAGCACGAGTCATC-3′), *COL1* (5′-GAGGGCCAAGACGAAGACATC-3′, 5′-CAGATCACGTCATCGCACAAC-3′), *DAGLA* (5′-TGAAATTATTCCTGCAAGCCAA-3′, 5′-CAGACATCTCTTCTCACCCTTCTTT-3′), *DAGLB* (5′-TCAGGTGCTACGCCTTCTC-3′, 5′-TCACACTGAGCCTGGGAATC-3′), *NAPEPLD* (5′-ACTGGTTATTGCCCTGCTTT-3′, 5′-AATCCTTACAGCTTCTTCTGGG-3′), *MGLL* (5′-GGAAACAGGACCTGAAGACC-3′, 5′-ACTGTCCGTCTGCATTGAC-3′), and *FAAH* (5′-CACACGCTGGTTCCCTTCTT -3′, 5′-GGGTCCACGAAATCACCTTTGA-3′). The expression levels of all target genes were normalized to the housekeeping gene *RPLP0* (5′-CTTCCTTAAGATCATCCAACTA-3′, 5′-ACATGCGGATCTGCTGCA-3′).

### 4.6. Western Blotting

Kidney homogenates were prepared using the BulletBlender^®^ and zirconium oxide beads (Next Advanced, Inc., Troy, NY, USA) in a RIPA buffer containing 25 mM Tris-HCl pH 7.6, 150 mM NaCl, 1% NP-40, 1% sodium deoxycholate, and 0.1% SDS. Protein concentrations were measured using the Pierce™ BCA Protein Assay Kit (Thermo Scientific, Hanover Park, IL, USA). Samples were separated by SDS-PAGE on 4–15% acrylamide gels at 150 V and transferred to nitrocellulose membranes using the Trans-Blot^®^ Turbo™ Transfer System (Bio-Rad, Hercules, CA, USA). Membranes were then blocked for 1 h in 5% milk (in 1x TBS-T) to prevent unspecific binding and incubated overnight at 4 °C with CB1R (ImmunoGenes, Budakeszi, Hungary, Cat #CB1), fatty acid amide hydrolase (FAAH; Abcam, Cambridge, UK, Cat #ab54615), and kidney injury marker 1 (KIM-1; Abcam, Cambridge, UK Cat #ab78494) antibodies. After washing, membranes were incubated with anti-rabbit or mouse horseradish peroxidase (HRP)-conjugated secondary antibodies for 1 h at room temperature, and chemiluminescence detection was performed using Clarity™ Western ECL Blotting Substrate (Bio-Rad, Hercules, CA, USA). Densitometry was quantified using ImageJ software (version 1.53k), and quantification was normalized to anti-β actin (Abcam, Cambridge, UK, Cat# ab49900) and VDAC (Abcam, Cambridge, UK, Cat #ab15895) antibodies.

### 4.7. Histopathology

Paraffin-embedded kidney sections (4 µm) were stained with H&E and Trichrome (Abcam, Cambridge, UK, Cat #ab150686). Images of both staining samples were captured from 10 randomly selected 20× fields using an AxioCam ICc5 color camera mounted on an Axio Scope.A1 light microscope (Zeiss, Oberkochen, Germany). Quantification of glomerular and Bowman’s space cross-sectional areas in H&E, as well as the quantification of the collagen cross-sectional areas in Trichrome staining, was carried out in a blinded manner using Adobe Photoshop CS3 software (version CS3). Bowman’s space area was normalized to the glomerular area to account for variations in glomerular size.

### 4.8. Statistical Analysis

Statistical analysis was conducted using SPSS version 26 and GraphPad Prism version 9. Fisher’s exact test was employed to assess the association between nominal variables in the obese and lean groups. Demographic and biochemical continuous variables are presented as median (range), and all other continuous variables are presented as mean ± SD. Differences between two groups were evaluated using the non-parametric Mann–Whitney test, with statistical significance set at *p* < 0.05. The Pearson correlation coefficient analysis was used to assess correlations between metabolic parameters and eCB levels, with statistical significance set at *p* < 0.05.

In our study, we harnessed the power of correlation matrices to uncover connections among the diverse variables we investigated. These matrices provide a foundational framework for comprehending the interdependencies between different factors. Correlation matrices were generated using the MATLAB version: 9.10.0.1851785 (R2021a) “corrcoef” function. This function facilitates the calculation of Pearson correlation coefficients, which serve as numeric indicators of the strength and direction of linear relationships between pairs of variables (positive correlations were displayed in blue, while negative correlations were depicted in brown). These coefficients are commonly used to quantify how closely two variables move together and to assess the degree of their association, thereby quantifying the extent of relationships between pairs of variables. This approach provided us with a quantifiable measure of how variables interact, offering insight into the degree and nature of their associations. By setting the “Rows” option to “pairwise”, we ensured that only complete data entries were used for comparisons, effectively preventing the influence of missing values.

Prior to analysis, the data underwent Z-score normalization. This technique adjusts the values of each feature to have an average of 0 and a standard deviation of 1. Such normalization ensures that various variables can be fairly compared and placed on the same scale, ultimately enhancing the accuracy of our subsequent analyses.

We conducted matrix analysis for all patients as well as for lean and obese patient subgroups, aiming to explore potential variations in the relationships between different factors. This method facilitated a comprehensive investigation of the intricate associations between various clinical and biochemical parameters, ultimately leading to a deeper understanding of their interplay.

## 5. Conclusions

In summary, our findings highlight the intricate connections between the renal ECS and systemic metabolic parameters. The observed relationships between renal eCBs, blood glucose levels, lipid profiles, and liver enzymes provide novel insights into the complex interplay of the ECS and metabolic homeostasis. Further investigations are warranted to unravel the underlying mechanisms and elucidate the potential implications of these findings in the context of metabolic disorders and renal health.

## Figures and Tables

**Figure 1 ijms-24-13636-f001:**
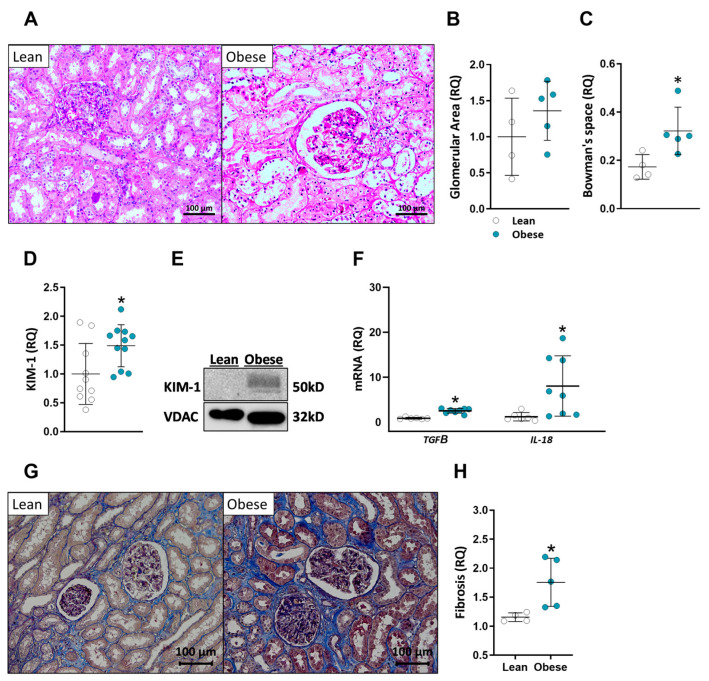
Kidney pathology of lean vs. obese individuals. (**A**) Representative images of ×20 magnified H&E-stained kidneys. Quantification of glomerular area (**B**) and Bowman’s space area (**C**). Kidney injury marker-1 (KIM-1) was measured via Western blot and quantified (**D**,**E**). *TGFB* and *IL-18* relative kidney mRNA expression (**F**). Representative images of ×20 magnified Trichrome-stained kidney sections (**G**) and fibrosis quantification (**H**). Data are mean ± SD; * *p* < 0.05 obese vs. lean group.

**Figure 2 ijms-24-13636-f002:**
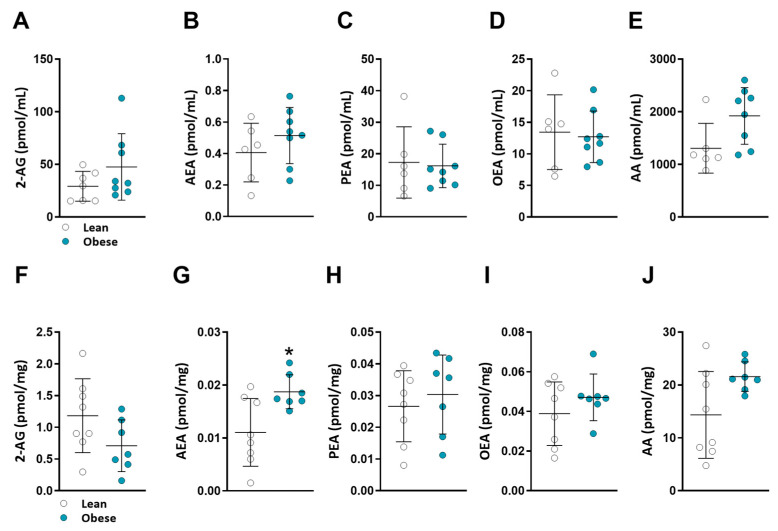
Serum and kidney eCB levels in lean vs. obese patients. 2-AG (**A**), AEA (**B**), PEA (**C**), OEA (**D**), and AA (**E**) levels were extracted from serum, and 2-AG (**F**), AEA (**G**), PEA (**H**), OEA (**I**), and AA (**J**) levels were extracted from kidney tissue. All eCBs were measured using LC-MS/MS. Data are mean ± SD; * *p* < 0.05 vs. lean group.

**Figure 3 ijms-24-13636-f003:**
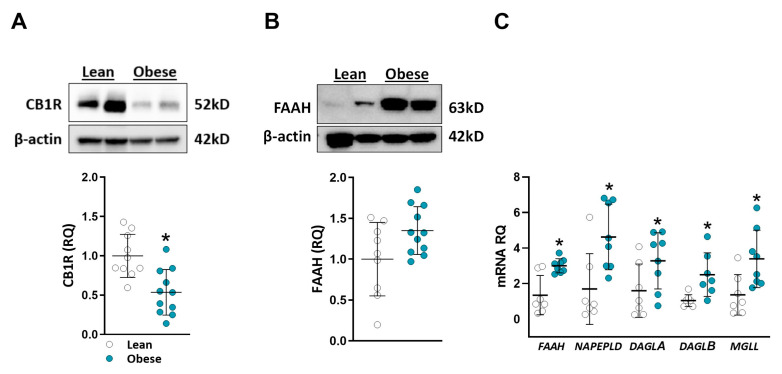
Kidney CB1R and ECS enzyme expression levels. Kidney proteins were extracted; CB1R (**A**) and FAAH (**B**) levels were examined via Western blot and quantified. Relative mRNA expression of *FAAH*, *NAPEPLD*, *DAGLA*, *DAGLB*, and *MGLL* genes (**C**). Data are mean ± SD; * *p* < 0.05 vs. lean group.

**Figure 4 ijms-24-13636-f004:**
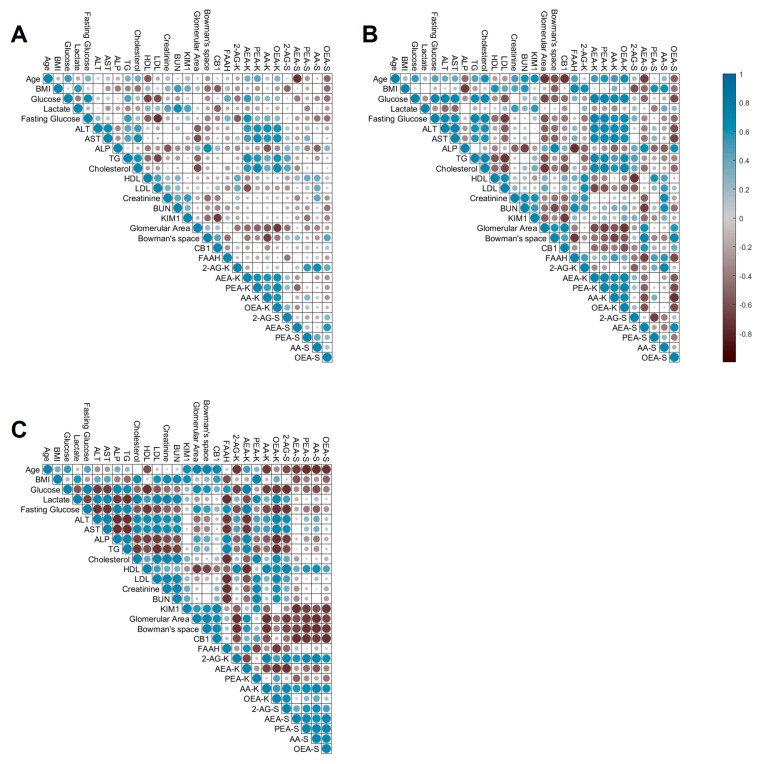
Large-scale correlation analysis. This correlation matrix displays the Pearson correlation coefficients between multiple variables. The matrix was applied to variables of all patients (**A**) and then stratified according to their BMI, analyzing the lean (**B**) and obese (**C**) groups separately. In eCBs measurements, K = kidney, and S = serum.

**Table 1 ijms-24-13636-t001:** Patient demographics and key parameters.

Parameters	Low BMI (n = 10)	High BMI (n = 11)	*p* Value
Age (years)	57.3 (40–64)	54.91 (38–71)	0.5214
BMI (kg/m^2^)	23.18 (19.02–25.99)	33.69 (30.02–42.50)	**<0.0001**
Fasting serum glucose (mg/dL) #	157 (75-362)	122.86 (89–199)	0.7396
Creatinine (mg/dL) *	1.41 (0.56–4.73)	0.78 (0.64–1.00)	0.7546
BUN (mg/dL) *	14.07 (9.07–31.09)	15.37 (10.67–24.22)	0.2414
Glucose (mmol/L) *	5.87 (3.99–6.67)	6.57 (3.34–9.97)	0.4908
Lactate (mmol/L) *	2.55 (1.13–4.53)	3.86 (2.3–6.5)	0.1079
ALT (U/L) *	18 (9–29)	18.87 (10–29)	0.323
AST (U/L) *	17.11 (9.2–24.1)	13.56 (9.2–18.2)	0.5987
ALP (U/L) *	90.97 (61.1–136.2)	74.33 (54.9–119.3)	0.1812
TG (mg/dL) *	182.55 (119.58–852.44)	214.33 (100.81–609.23)	0.4908
Cholesterol (mg/dL) *	164.91 (112.26–221.72)	151.21 (111.4–218.61)	0.5728
HDL (mg/dL) *	0.98 (0.63–1.13)	0.91 (0.61–1.36)	0.8258
LDL (mg/dL) *	70.33 (18.10–106.95)	69.51 (1.63–139.25)	0.8765

* Measured from 15 patients due to the lack of serum samples. # Presented from 17 patients due to the lack of data. Bold highlights the significant change between the two groups.

**Table 2 ijms-24-13636-t002:** The collision energy (CE), declustering potential (DP), and collision cell exit potential (CXP) for the measured eCBs.

Analyte	Molecular Ion [M + H]^+^ [M − H]^−^ for AA [*m*/*z*]	Fragment (*m*/*z*)	DP (Volts)	CE (Volts)	CXP (Volts)
2-AG	379.2	287.1 (quantifier)	70	19	14
		91 (qualifier)	70	67	10
AEA	348.2	287.1 (quantifier)	26	13	16
		62 (qualifier)	26	13	8
PEA	300.3	283.2 (quantifier)	130	19	24
		62 (qualifier)	130	17	8
AA	305.3	91 (quantifier)	1	49	10
		287.1 (qualifier)	1	13	22
OEA	326.3	61.9 (quantifier)	146	21	24
		309.1 (qualifier)	146	21	42
d_4_-AEA	352.3	287.1 (quantifier)	66	15	20
		66 (qualifier)	66	21	8

## Data Availability

The entire data set presented in this study is available within the article.

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
