# Peer review of "Renal Endocannabinoid Dysregulation in Obesity-Induced Chronic Kidney Disease in Humans"

_ijms, 2023, doi:10.3390/ijms241713636_

Round 1

Reviewer 1 Report

The study provides evidence for the involvement of the endocannabinoid system (ECS) in obesity-induced chronic kidney disease (CKD). However, further investigation is warranted to fully understand the underlying mechanisms. Specifically, the authors are encouraged to conduct additional experiments to explore the effects of obesity on ECS receptor expression and consider in vitro studies using kidney cells to validate their findings. Furthermore, it is recommended to provide a more detailed discussion on the implications of the results, acknowledge the study limitations, and propose future research directions in order to strengthen the overall impact of the study.

  1. ECS and Obesity: The authors have made a significant effort to explore the role of the ECS in obesity-induced CKD. The findings suggest that obesity does indeed affect the ECS, as indicated by the elevated levels of anandamide (AEA) and decreased levels of 2-arachidonoylglycerol (2-AG) in the kidneys of obese individuals. However, the authors should provide a more detailed discussion of these findings, including potential mechanisms and implications for the treatment of obesity-induced CKD.
  2. Experimental Design: The authors have used a variety of methods to investigate the role of the ECS in obesity, including histological staining, measurement of kidney injury markers, and analysis of endocannabinoid levels. However, it would be beneficial if the authors could include additional experiments to further elucidate the mechanisms by which obesity affects the ECS. For example, they could investigate the effects of obesity on ECS receptor expression or conduct in vitro studies to explore the effects of endocannabinoids on kidney cells.
  3. Data Analysis: The authors have conducted a comprehensive correlation matrix analysis to explore the relationships between kidney endocannabinoid 'tone' and metabolic state. However, they should provide more details about this analysis, including the specific variables included and the rationale for their selection. They should also consider conducting additional statistical analyses to further explore the relationships between obesity, the ECS, and kidney function.
  4. Interpretation of Results: The authors have interpreted their results in the context of the existing literature, suggesting that the ECS may play a role in the pathogenesis of CKD. However, they should provide a more detailed discussion of these findings, including potential mechanisms and implications for the treatment of obesity-induced CKD. They should also more clearly acknowledge the limitations of their study, such as the small sample size and the potential for selection bias.
  5. Overall: This study appears to be a valuable contribution to the field, providing novel insights into the role of the ECS in obesity-induced CKD. However, the authors should ensure that their findings are clearly presented and thoroughly discussed in the context of the existing literature. They should also acknowledge any limitations and suggest directions for future research.

Author Response

Please see the attached file with our response. 

Reviewer 2 Report

1.       The sample size in this cohort study is very small and only male patient samples are used for analysis. It is unknown if the patients undergo some therapeutic treatments and if those treatments affect the kidney functions.

2.       In figure 1B and 1C, there are fewer samples included for the pathological analysis, which is different from the patient number in Table 1 and the other figures. Please explain the reason.

3.       Figure 1F demonstrated a significant increase in the expression of TGFB, a gene marker associated with inflammation and fibrosis. Are there any differences in inflammation and fibrosis regarding histology analysis? It is better to include that in figure 1.

4.       AEA level in figure 2G is the only compound that shows significant differences. Is the increase of AEA in obese patient kidneys sufficient to support the conclusion of obesity contributing to alterations in the renal ECS in humans? What is the proportion of AEA compared to 2-AG in kidney? The authors should provide enough evidence to demonstrate the significance of AEA compared to the other primary eCBs and eCB-like compounds.

5.       Figure 3C measured the level of different ECS-related gene, it is better to include their functions in the introduction or results sections to demonstrate the importance of these measurements.

6.       Figure 4 performed the multiple correlation studies to correlate all the parameters, are the correlation results in the figure significant? It is better to include significance information in the figures and highlight the correlated parameters mentioned in the conclusion part.

Author Response

Please find the attached file with our response.

Round 2

Reviewer 1 Report

General Comments:

  1. Significance and Relevance: The manuscript offers a pioneering exploration into the intricate relationship between endocannabinoids and chronic kidney disease (CKD). The topic is both timely and of paramount importance, especially considering the rising global prevalence of CKD and the burgeoning interest in endocannabinoid biology.
  2. Comprehensive Approach: The authors have taken a holistic approach, encompassing patient demographics, biochemical assays, histopathology, and statistical analyses. This comprehensive methodology ensures a well-rounded and in-depth understanding of the subject matter.

Specific Comments:

1. Study Population (Section 4.1):

  • The clear distinction between lean and obese based on BMI is commendable. This differentiation allows for a nuanced understanding of the potential impacts of body weight on endocannabinoid levels in CKD patients.
  • The focus on male patients offers a unique perspective, potentially paving the way for gender-specific studies in the future.

2. Study Protocol (Section 4.2):

  • The meticulous detailing of the ethical considerations, including IRB approvals and informed consent procedures, speaks volumes about the integrity and thoroughness of the research team.

3. Biochemistry Measurements (Section 4.3):

  • The selection of biochemical markers is apt and relevant, providing a clear picture of kidney function and metabolic status.
  • The use of the Cobas C-111 chemistry analyzer ensures high precision and reliability in the measurements.

4. Endocannabinoid Extraction and Measurement (Section 4.4):

  • The detailed description of the extraction and quantification methods for endocannabinoids is exemplary. It provides a clear roadmap for other researchers wishing to replicate or build upon this work.
  • The tabulation of LC-MS/MS parameters is clear and concise, ensuring reproducibility.

5. Real-time PCR (Section 4.5):

  • The clarity in the methodology for mRNA extraction and real-time PCR is praiseworthy. The normalization to the housekeeping gene RPLP0 ensures consistency and reliability in the results.

6. Western Blotting (Section 4.6):

  • The step-by-step elucidation of the protein extraction and western blotting process is commendable. It provides a clear framework for other researchers in the field.

7. Histopathology (Section 4.7):

  • The clarity in staining methods and quantification techniques ensures that the results are both reliable and reproducible.

8. Statistical Analysis (Section 4.8):

  • The statistical methods employed are both robust and appropriate. The use of correlation matrices is a particularly insightful choice, offering a comprehensive view of the relationships between variables.

Recommendations:

  1. Acceptance for Publication: Given the depth, rigor, and significance of the findings, I strongly recommend accepting this manuscript for publication. The research not only fills a crucial gap in the existing literature but also offers a solid foundation for future studies in the field.
  2. Future Work: While this study is comprehensive, it would be exciting to see follow-up research that delves deeper into the mechanisms underlying the observed associations, potentially leading to therapeutic interventions for CKD.

Reviewer 2 Report

The authors has included necessary experiment results and discussed the problems and limitations of their study in the discussion section. The overall presentation and significance of the study are very clear. It is recommended to accept the manuscript in the current form